# Genome assembly and chemogenomic profiling of National Flower of Singapore *Papilionanthe* Miss Joaquim 'Agnes' reveals metabolic pathways regulating floral traits

Abner Herbert Lim [1], Zhen Jie Low[1], Prashant Narendra Shingate[2], Jing Han Hong[1,3], Shu Chen Chong[1], Cedric Chuan Young Ng[1], Wei Liu[1], Robert Vaser[2,4], Mile Šikić[2,4], Wing-Kin Ken Sung[2,5], Niranjan Nagarajan [2,5], Patrick Tan[2,3,6,7✉] & Bin Tean Teh [1,2,3,7,8,9✉]

Singapore's National Flower, *Papilionanthe* (*Ple.*) Miss Joaquim 'Agnes' (PMJ) is highly prized as a horticultural flower from the Orchidaceae family. A combination of short-read sequencing, single-molecule long-read sequencing and chromatin contact mapping was used to assemble the PMJ genome, spanning 2.5 Gb and 19 pseudo-chromosomal scaffolds. Genomic resources and chemical profiling provided insights towards identifying, understanding and elucidating various classes of secondary metabolite compounds synthesized by the flower. For example, presence of the anthocyanin pigments detected by chemical profiling coincides with the expression of *ANTHOCYANIN SYNTHASE (ANS)*, an enzyme responsible for the synthesis of the former. Similarly, the presence of vandaterosides (a unique class of glycosylated organic acids with the potential to slow skin aging) discovered using chemical profiling revealed the involvement of glycosyltransferase family enzymes candidates in vandateroside biosynthesis. Interestingly, despite the unnoticeable scent of the flower, genes involved in the biosynthesis of volatile compounds and chemical profiling revealed the combination of oxygenated hydrocarbons, including traces of linalool, beta-ionone and vanillin, forming the scent profile of PMJ. In summary, by combining genomics and biochemistry, the findings expands the known biodiversity repertoire of the Orchidaceae family and insights into the genome and secondary metabolite processes of PMJ.

[1] SingHealth Duke-NUS Institute of Biodiversity Medicine, Singapore, Singapore. [2] Genome Institute of Singapore, A*STAR, Singapore, Singapore. [3] Duke-NUS Medical School, Singapore, Singapore. [4] Faculty of Electrical Engineering and Computing, University of Zagreb, Zagreb, Croatia. [5] School of Computing, National University of Singapore, Singapore, Singapore. [6] Cancer Science Institute of Singapore, National University of Singapore, Singapore, Singapore. [7] SingHealth/Duke-NUS Institute of Precision Medicine, Singapore, Singapore. [8] Institute of Molecular and Cell Biology, Singapore, Singapore. [9] National Cancer Center Singapore, Singapore, Singapore. ✉email: tanbop@gis.a-star.edu.sg; teh.bin.tean@singhealth.com.sg

**B**red in 1893 by Miss Agnes Joaquim, *Papilionanthe (Ple.)* Miss Joaquim 'Agnes' (PMJ), commonly known as *Vanda* Miss Joaquim, was highly prized as a horticultural flower in the early 19th Century in Hawaii and many parts of Europe. The flower was crowned the National Flower of Singapore on 15 April 1981[1]. PMJ is a hybrid cross between *Ple. Teres* and *Ple. hookeriana* (Fig. 1a), where its parentage was ascertained through molecular evidence derived from the *matK* and *rbcL* regions of the chloroplast genome[2]. PMJ growth occurs continuously in a monopodial fashion at the stem tip. The leaves are terete (Fig. 1b) and grow at the apex. The inflorescence period typically starts at the height of 1.4–1.5 m[3]. The flower has three sepals (one dorsal and two lateral) and three two rosy-violet lateral petals, and a modified labellum with a large and broad lip that is brightly purple with an orange center. The flower can grow up to 6 cm upon maturity (Fig. 1a).

Orchids harbor considerable amounts of therapeutic bioactive compounds with ethnobotanical uses in traditional pharmacopeias[4,5]. To date, a wide variety of bioactive compounds with diverse pharmacological effects, such as flavonoids, phenylpropanoid derivatives, terpenoids, and vandaterosides of glucopyranosyloxybenzyl eucomate derivatives, have been isolated and extracted from various organs of orchids such as leaves, stems, roots, and flowers[6]. However, these phytochemicals represent a small subset of the natural bioactive produced by orchids, with many more to be discovered[7]. Hence, multi-omics profiling leading towards an understanding of the biosynthesis of secondary metabolites and phytochemicals is essential for finding novel phytochemicals and subsequent studies of their health-promoting properties and related risks such as toxicity[8,9]. The genomes of several Orchidaceae species such as *Apostasia*[10], *Dendrobium*[11–14], *Gastrodia*[15–17], *Phalaenopsis*[18,19], and *Vanilla*[20] have been sequenced for developmental biology studies and to uncover biosynthetic pathways of phytochemicals that regulate horticultural floral features, and compounds which are anti-inflammatory, anti-microbial and anti-oxidative in nature.

In this study, the integration of genomic and chemical profiling was exploited to to investigate phytochemicals produced by PMJ. The genome of PMJ was assembled to understand specific genes and pathways involved in producing secondary metabolites, serving as a guide to identify these compounds through chemical profiling. This integrative approach empowers a comprehensive understanding of pigmentation, biosynthesis of potential therapeutic metabolites and scent in PMJ. As a promising alternative

to conventional chemical profiling and bioactivity screening strategies, this integrative approach may further contribute to understanding the rich repertoire of Orchidaceae biodiversity.

## Results

**Genome assembly and annotation of *Ple*. Miss Joaquim 'Agnes'.** A reference genome of PMJ (Fig. 2) was assembled de novo by integrating long-range sequencing reads from Oxford Nanopore Technology (ONT) and polished using Illumina short-reads from whole-genome shotgun sequencing to produce the initial assembly. Hi-C chromatin interaction data was then used for scaffolding, correcting misjoins, and reorientating the assembled sequences into 19 pseudo-chromosomal scaffolds (Supplementary Data 1). This observation was concordant with an earlier study showing a diploid chromosome of $2n = 38$ in hybrid crossing between *Ple. Hookeriana* x *Ple. Teres*[21,22]. Similarly, $2n = 38$ chromosomes are commonly reported across species of the Orchidaceae family, such as *Dendrobium officinale*[11] and *Phalaenopsis equistris*[18,19]. The genome size of PMJ was estimated to be 2 Gb based on k-mer distribution analysis using GenomeScope[23] (Supplementary Fig. 1a). The eventual genome assembly (Supplementary Fig. 1b) used for downstream analysis was 2.5 Gb with an N50 of 159.5 Mb and a GC content of 35.3% (Supplementary Data 2).

Evaluation of assembly completeness using Merqury[24] revealed a Quality Value (QV) of 27, and BUSCO[25] assessment recovered 1543 out of 1614 (95.6%) conserved genes from the Viridiplantae lineage (Supplementary Data 2). Repetitive elements within the PMJ genome were annotated for low complexity sequences and interspersed repetitive DNA using RepeatMasker[26], supplied with a library of transposable element (TE) families discovered using RepeatModeler[27]. The annotated repetitive elements occupy 78% of the PMJ genome consisting of interspersed repeats containing retroelements (53%), long terminal repeat (LTR) elements (48%), and DNA transposons (1.52%) as well as simple (1.56%) and low complexity (0.13%) repeats (Supplementary Data 3). A total of 31,529 protein-coding genes were annotated by reconciling evidence from transcripts and proteins with a mean of 5.3 exons per gene. BUSCO assessment of annotated genes recovered 91.2% of core genes from the viridiplantae lineage (Supplementary Data 2), with functional assignment across 87% of genes identified.

*Phylogeny inference across representative plant species.* Reference genomes from 15 representative plant species comprising 10 monocots[28–32], including orchid genomes such as *Phalaenopsis*[19], *Vanilla*[20], *Dendrobium*[12–14], and 5 eudicots[33–36] (Fig. 3a and Supplementary Data 4), were downloaded to determine phylogenetic relationships across these species. A total of 32,238 orthologous gene clusters were identified using Orthofinder. 7666 (23%) orthogroups were shared across all species. 104 single-copy orthogroups from 15 individual species were identified. Phylogenetic analysis was conducted using translated protein sequences selected from the identified 104 single-copy orthogroups. PMJ shared phylogenetic resemblance with *P. equistris* and formed a clade with species from the Orchidaceae family (Fig. 3a).

*MADS-box TF profile during mature blooms.* MADS-box genes are highly involved in plant growth and development, ranging from vegetative to reproductive phases [37,38]. To understand the role of MADS-box transcriptional factors (TFs) in the leaf and blooms of PMJ, a Hidden Markov Model (HMM) search was performed on the Pfam[39] SRF Type transcription factor (SRF-TF: PF00319) HMM model. 67 MADS-box TFs were identified within the PMJ genome (Supplementary Data 5). The MADS-box TFs

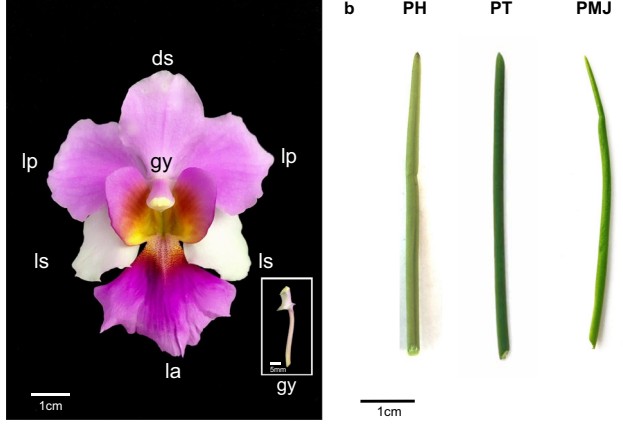

**Fig. 1 *Ple*. Miss Joaquim 'Agnes'. a** The flower of *Ple*. Miss Joaquim 'Agnes' in front view with a dorsal sepal (ds), two lateral petals (lp), labellum (la), a modified petal, two lateral sepals (ls) and the gynostemium (gy), which is enlarged in the white-bordered inset image. **b** Terete leaves of *Ple. hookeriana* (PH), *Ple. Teres* (PT) and the hybrid *Ple*. Miss Joaquim 'Agnes' (PMJ), from left to right.

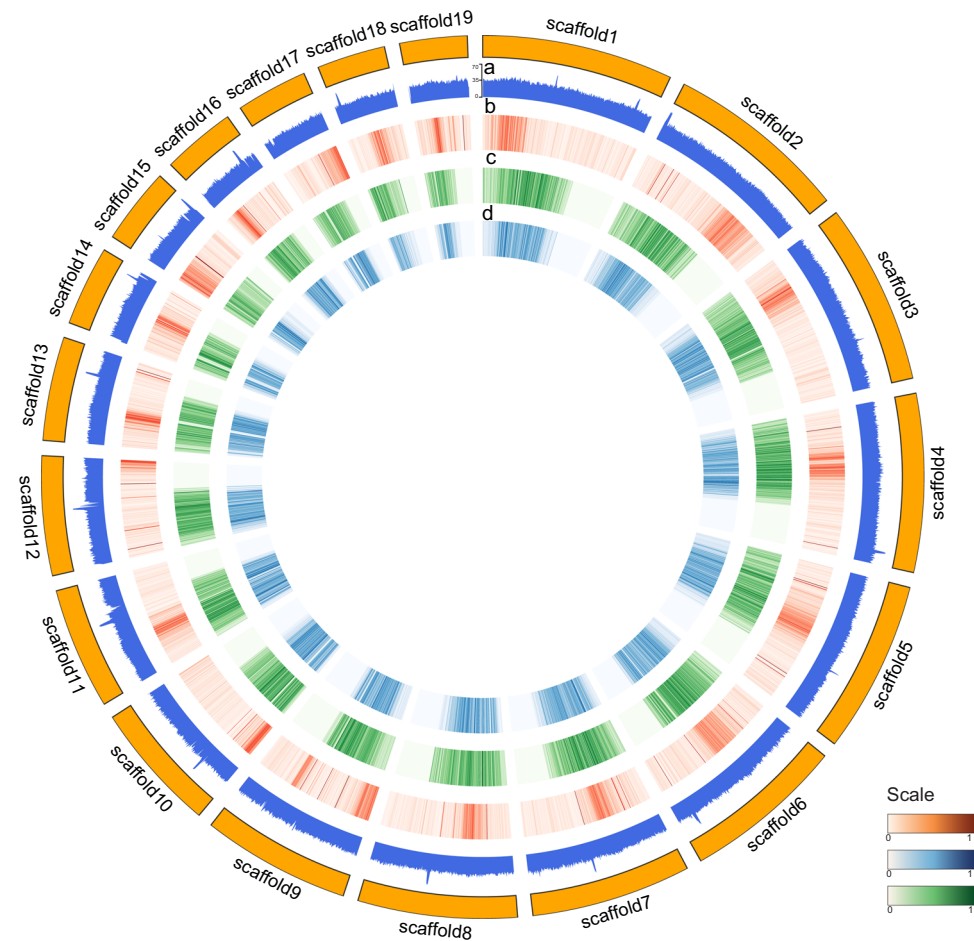

**Fig. 2 Circos representation of *Ple*. Miss Joaquim 'Agnes'.** A circos plot of genomic features across 19 pseudo-chromosomal assembled scaffolds, from the outer track to the inner track, **a** filled line plot of GC content, and the following heatmap shows **b** gene density, **c** LTR/Copia density and **d** LTR/gypsy density across the assembled scaffolds.

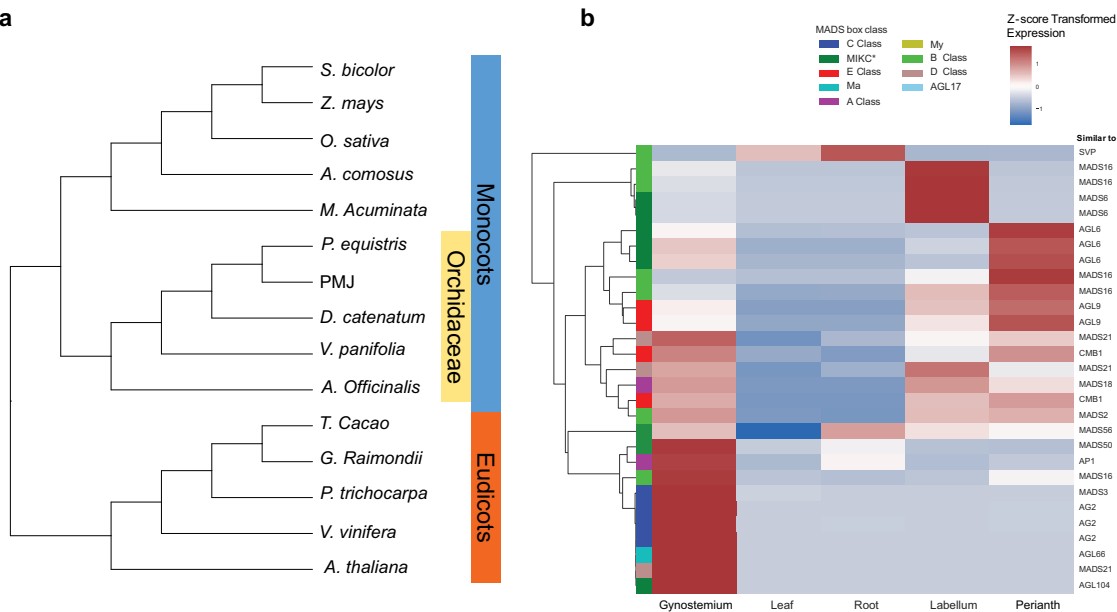

**Fig. 3 Phylogeny and MADS-box TFs *Ple*. Miss Joaquim 'Agnes'.** **a** Phylogenetic relation between selected taxa within the Orchidaceae family, selected monocots, and eudicots. **b** Expressed MADS-Box gene TFs across flowering and vegetative parts of PMJ during mature bloom.

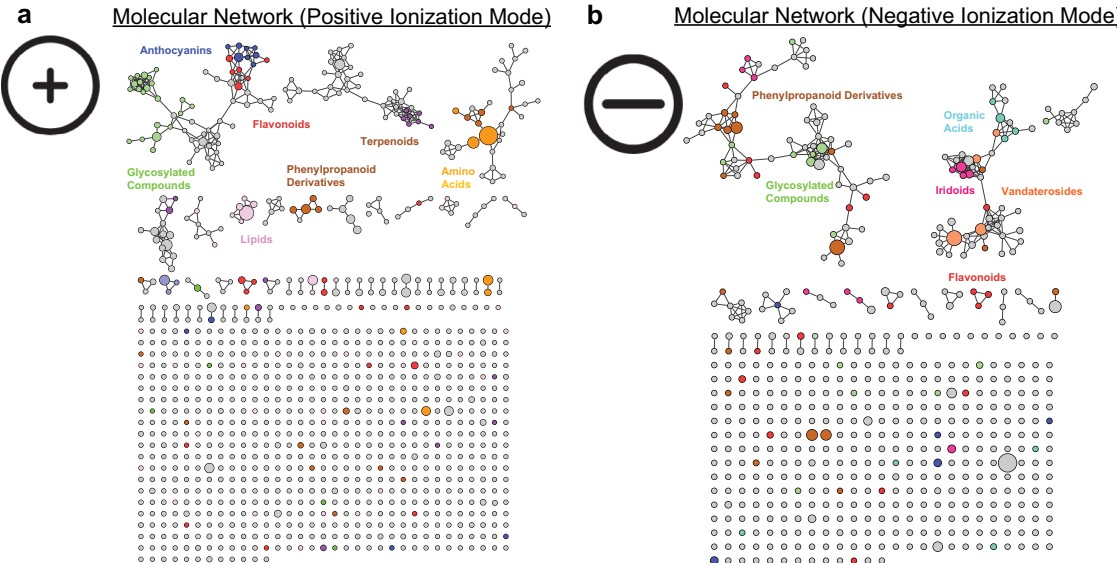

**Fig. 4 Manually curated molecular network of 1500 chemical features discovered from tissues of the flower and leaves of PMJ. a** LC-HRMS in positive ionization mode. **b** LC-HRMS in negative ionization mode.

were subsequently identified and classified into their subclasses through codon alignment and phylogeny inference using homologous MADS-box class proteins from *O. sativa, A. thaliana, D. catenatum* and *P. equistris* (Supplementary Data 5). Transcriptome analysis of the mature bloom of PMJ, leaves and roots displayed enhanced expression levels of *MADS16* and *MADS6* (Fig. 3b) in the flower's labellum. This elevated expression shows the involvement of *MADS6* and specialized *MADS16* in the regulation of the mature labellum while Class-C MADS-box TFs are involved in the regulation of the gynostemium. In PMJ, MADS-box genes have specialized functions across mature tissues participating in various regulatory processes.

*Transcriptomics analysis from PMJ tissues.* Tissue-specific expression across the various floral and leaf tissues of PMJ was investigated using RNA-seq from different floral and leaves tissues with a minimum of two biological replicates. Gene ontology enrichment was performed with clusterProfiler[40] using differentially expressed genesets from RNA-seq to understand the tissue-specific molecular function and biological processes. Principal component analysis and hierarchical clustering across the spectrum of tissues profiled (Supplementary Fig. 2) confirmed the variability in gene expression across the tissue types. However, it was evident that in both hierarchical clustering and principal component analysis, the petal and sepals were clustered similarly. Hence sepal/petal (perianth) are grouped in subsequent analysis.

Pairwise differential expression was performed using DESeq2[41] across all tissue combinations, and upregulated genes were compared to identify tissue-specific expressions. Using a Log2FoldChange >1 as cutoff, 526 genes were found upregulated only in leaves, 64 genes were upregulated only in the labellum, 516 genes were determined to be upregulated only in the root, 65 genes were upregulated only in perianth tissues, and 118 genes were found to be involved in the regulation of the gynostemium. Enrichment of GO terms was used to understand the biological pathways significant across tissue-specific genesets.

GO pathways enriched in the leaves of PMJ revealed activities related to photosynthesis, chloroplast activity and precursors for generating energy/metabolites. In the roots, GO terms enriched mainly belong to response to water and production of secondary metabolites, which could play crucial roles in the uptake of nutrients and water from the air and their surroundings

(Supplementary Fig. 3). In the floral tissues, gene-set enrichment specific to the perianth shows high specificity towards the regulation of anthocyanin-based metabolism mainly involved in floral pigmentation. GO terms enriched in labellum tissues shows gene-set enrichment involved in the biosynthesis of many secondary metabolites, which could be used to attract pollinators and GO terms enriched in the gynostemium show processes involved in regulating floral development (Supplementary Fig. 4). Therefore, the identification of differentially expressed genes and their enriched pathways demonstrates the diverse biological processes occurring in the various tissues of PMJ that allow each tissue to play specialized roles in growth and development.

**Analysis of Secondary Metabolite Biosynthesis in *Ple*. Miss Joaquim 'Agnes'.** Secondary metabolite profiling of PMJ revealed 253 compounds (66 volatile compounds, 187 non-volatile compounds). Profiling was performed using gas chromatography high-resolution mass spectrometry (GC-HRMS) and liquid chromatography high-resolution mass spectrometry (LC-HRMS) for volatile and non-volatile compounds, respectively. Non-volatile compound profiling of the plant extracts using LC-HRMS in both positive and negative ion modes revealed a collection of 1500 chemical features (Fig. 4 and Supplementary Fig. 5). Approximately 14% of these chemical features were manually curated to uncover the pathways governing the biosynthesis of these phytocompounds.

*Role of anthocyanins in floral color pigmentation.* The anthocyanin biosynthesis pathway, the extension of the general flavonoid pathway, is pivotal in the production of the wide distribution of natural color pigments ranging from pink to purple, responsible for orchid floral color formation[37,42–44]. Functional annotation of the discovered gene models revealed the presence of 62 enzymes with roles in the anthocyanin biosynthesis process in the PMJ (Supplementary Data 6). Transcriptomic analysis of the various PMJ tissues highlights the higher expression of *FLAVANONE 3-HYDROXYLASE (F3H)*, which regulates flavonoid biosynthesis, in the perianth (petal/sepal), labellum and gynostemium tissues compared to the average of other tissues (Log2FC: 4.7, padj: 3.12E-17). Downstream, *ANTHOCYANIN SYNTHASE (ANS)*, which governs the production of anthocyanins[45], which are

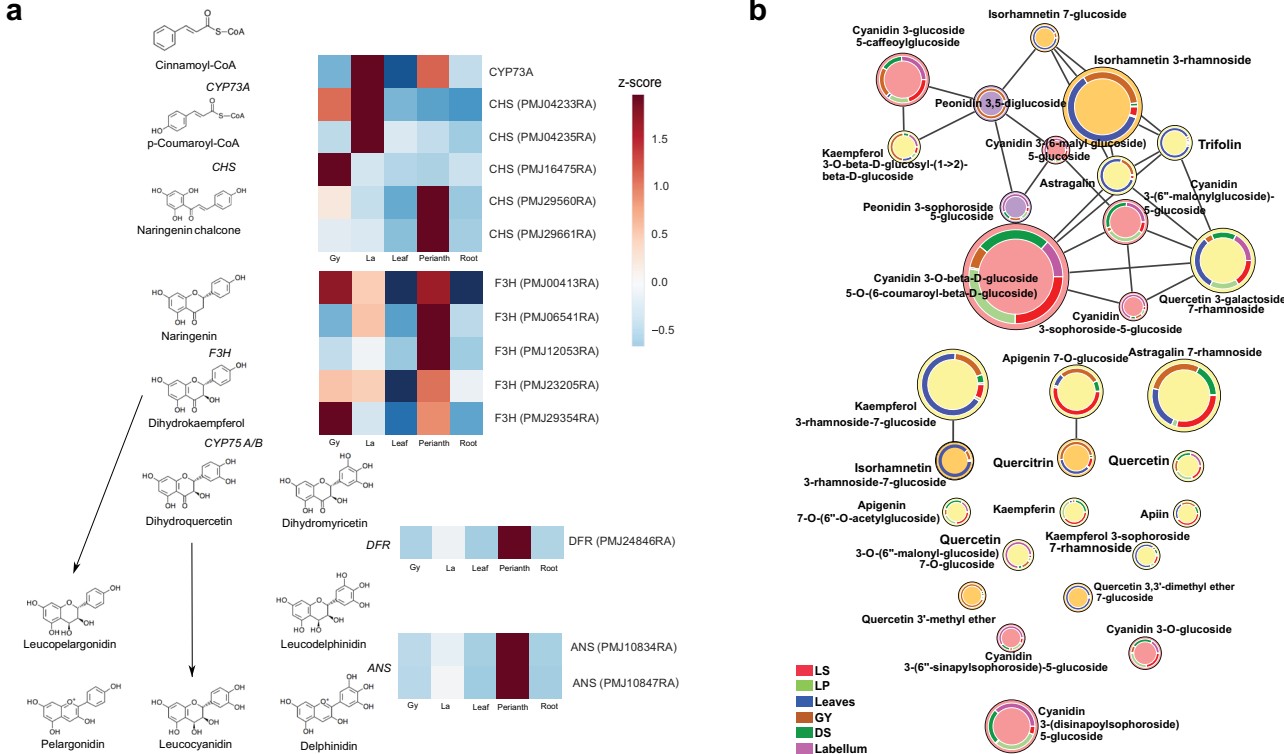

**Fig. 5 Anthocyanin biosynthesis pathway and the distribution of color pigments from measured metabolite profile. a** Enzymes involved in the anthocyanin biosynthesis pathway with a heatmap of their relative expression (with z-score transformation) across the flower and leaves of PMJ. **b** Network of flavonoid and anthocyanidin derivatives based from GNPS. Each node represents a derivative compound and the size illustrates the relative total ion abundance across the various PMJ tissues. (Source data: Supplementary Data 15, LS lateral sepal, LP lateral petal, GY gynostemium, DS dorsal sepal).

precursors for the production of anthocyanin-based pigments in flowers, were found to have a higher expression (Log2FC: 6, padj: 1.46E-25) in perianth (petal/sepal) and labellum tissue compared to other profiled tissues (Fig. 5a). These suggest that anthocyanin-based pigments are present in these floral tissues.

Non-volatile profiling using LC-HRMS revealed the interaction between 31 compounds from the anthocyanin biosynthesis pathway and identified their varied abundance based on fragment similarity (Fig. 5b). Amongst anthocyanin derivatives (Supplementary Fig. 6a), peonidin and cyanidin contribute to magenta and reddish-purple color respectively at neutral pH. In contrast, kaempferol, apigenin and quercetin derivatives give rise to a yellow-orange hue[46–48]. The color intensity contributed by various anthocyanin derivatives in the flower and leaf PMJ was consistent with the ion abundance of the respective anthocyanin compound from LC-HRMS profiling (Supplementary Fig. 6b). This observation is also consistent with the reddish-purple observed pigment phenotype of the flower (Fig. 1a). Overall, among the flower parts profiled using LC-HRMS (Supplementary Fig. 6b), the highest abundance of anthocyanins was found in the labellum (~1.34 e7), followed by dorsal sepal and lateral petal (both are ~1.17 e6) and lastly, gynostemium (~9.0 e6).

*Vandaterosides in flowers and leaves.* Vandaterosides, which are glucosyloxybenzyl eucomate derivatives, were identified in the leaf and floral tissues of PMJ using LC-HRMS. Vanderosides are natural compounds with the potential to remedy skin aging first discovered in the leaves and stems of *Ple. Teres*[49]. Deconstruction of vandaterosides reveals the presence of glucose, cinnamic acid, and p-hydroxybenzyl alcohol, making it reasonable to

postulate the involvement of acyltransferases and glucosyltransferases during their biosynthesis. Eucomic acid, which is likely formed through the esterification of eucomic acid in the presence of p-hydroxybenzyl alcohol, is the required precursor before vandaterosides synthesis[50]. Homologs of *ACYL-COA DEPENDENT ACYLTRANSFERASE* (*BAHD-AT*) and *ACYL-GLUCOSE DEPENDENT ACYLTRANSFERASE* (*SCPL-AT*) encode for enzymes that serve as catalysts for the esterification of eucomic acid during biosynthesis[51,52]. Both types of acyltransferase function through different mechanisms and are localized in separate intracellular compartments, with *BAHD-ATs* predominantly localized in the cytosol while *SCPL-ATs* are commonly found in vacuoles[50–53]. Subsequent glucosylation of the p-hydroxybenzyl moiety by a *UDP-sugar dependent glycosyltransferase (UGT)* [54] *or acyl-sugar dependent glucosyltransferase (GH1-GT)*[54,55] completes the biosynthesis of vandateroside I. The formation of vandateroside II follows a similar biosynthetic route as vandateroside I, albeit on the other free carboxyl moiety of vandateroside I. Further derivation into vandateroside III would involve another acyltransferase and glucosyltransferase.

A *BAHD* gene was found on the assembled PMJ genome (Fig. 6a) and could likely encode an enzyme to support the formation of eucomic acid in the presence of p-hydroxybenzyl alcohol. This precursor is then likely to be used by glucosyltransferase enzyme candidates from the flavonoid pathway (Supplementary Data 6, 7), which are abundant across the perianth tissues of PMJ, to synthesize vandaterosides. The localization of eucomic acid in the gynostemium was also unveiled through LC-HRMS, as well as abundant levels of vandateroside I, II and III picked up across the various tissues of PMJ (Fig. 6b).

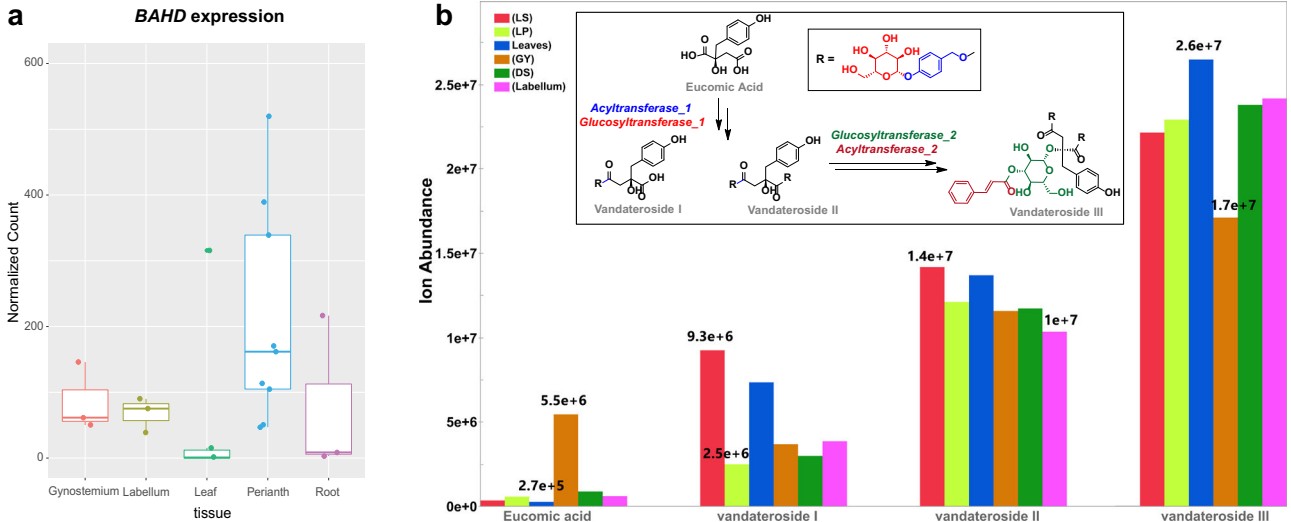

**Fig. 6 Putative biosynthetic pathway of Eucomic acids and their derivatives. a** Box plot showing the transcriptomic expression of *BAHD* gene across the various plant tissues of PMJ. For each tissue, $n = 3$ biological independent samples were considered center lines show the medians and box limits indicating the 25th and 75th percentiles. Wilcoxon rank sum test was performed pairwise across tissue groups; no significant differences across means are observed. (Source Data: Supplementary Data 16, Supplementary Data 17, *$p <= 0.05$, **$p < 0.01$). **b** Putative biosynthetic pathway of Eucomic acids and their derivatives Vandateroside I, II and III. The ion abundance of eucomic acids and their derivatives was detected using LC-HRMS across floral and leaf tissues. (Source Data: Supplementary Data 18, LS lateral sepal, LP lateral petal, GY gynostemium, DS dorsal sepal).

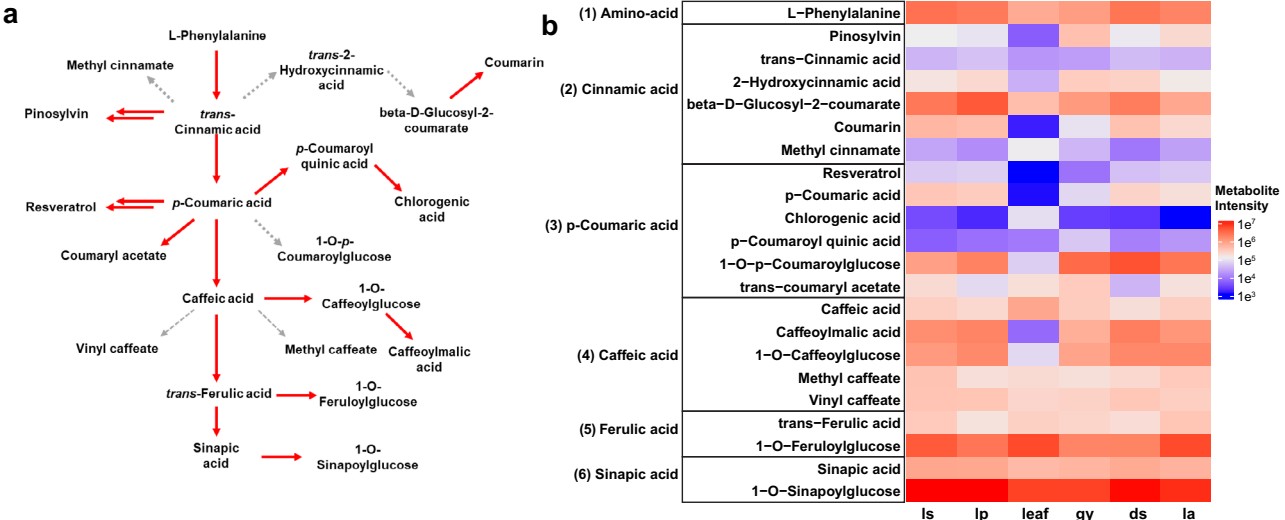

**Fig. 7 Phenylpropanoid and stilbenoid derivatives identified in *Ple*. Miss Joaquim 'Agnes'. a** Phenylpropanoid and stilbenoid biosynthesis pathway and their secondary metabolites. **b** LC-HRMS measured the metabolite intensity of phenylpropanoid and stilbenoid derivatives across leaf tissues of PMJ.

*Phenylpropanoids and Stilbenoids.* Phenylpropanoids are a diverse group of natural products synthesized by the deamination of phenylalanine into trans-cinnamic acid[56] (Fig. 7a). Trans-cinnamic acid is a precursor to biosynthesize many downstream metabolites such as caffeic acid, trans-ferulic acid and sinapic acid. Further glucosylation of these phenolic acids occurs through a *UDP-glucose glucosyltransferase* [EC 2.4.1.120][54] with substrate preference for hydroxycinnamic acids to yield glucose esters such as 1-O-p-coumaroylglucose, 1-O-caffeoylglucose, 1-O-feruloylglucose, and 1-O-sinapoylglucose. These glucose esters were identified in the extracts of PMJ using LC-HRMS (Fig. 7a and Supplementary Data 7) and serve as substrates for acylation or glucosylation of other secondary metabolites by *SCPL-ATs* or *GH1-GTs*, respectively. In general, the production of phenolic acids ubiquitously as a diversification strategy to biosynthesize many other downstream secondary metabolites[57]. Phenylpropanoid is synthesized from

phenolic acids through a series of enzymatic reactions. Therefore, the low concentrations of caffeic acid (a phenolic acid), in contrast with the abundance of caffeoylmalic acid (a phenylpropanoid), are due to the absence of a *SERINE CARBOXYPEPTIDASE-LIKE 8* acyltransferase [EC 2.3.1.92] (Fig. 7b).

Similar to phenylpropanoids, stilbenoids are biosynthesized starting from the deamination of phenylalanine to trans-cinnamic acid (Fig. 7a). With the addition of an acetyl-CoA group to form cinnamoyl-CoA, stilbene synthases extend cinnamoyl-CoA or p-Coumaroyl-CoA with three malonyl-CoA substrate units to yield the eventual stilbene core[57,58]. Stilbene synthase genes [EC 2.3.1.95] (Supplementary Data 7) associated with the production of the two stilbenoids (i.e., pinosylvin and resveratrol) were identified in the genome of PMJ. These stilbenoids were also detected in floral tissues from LC-HRMS profiling (Fig. 7b). Pinosylvin production is mediated by stress and physical damage,

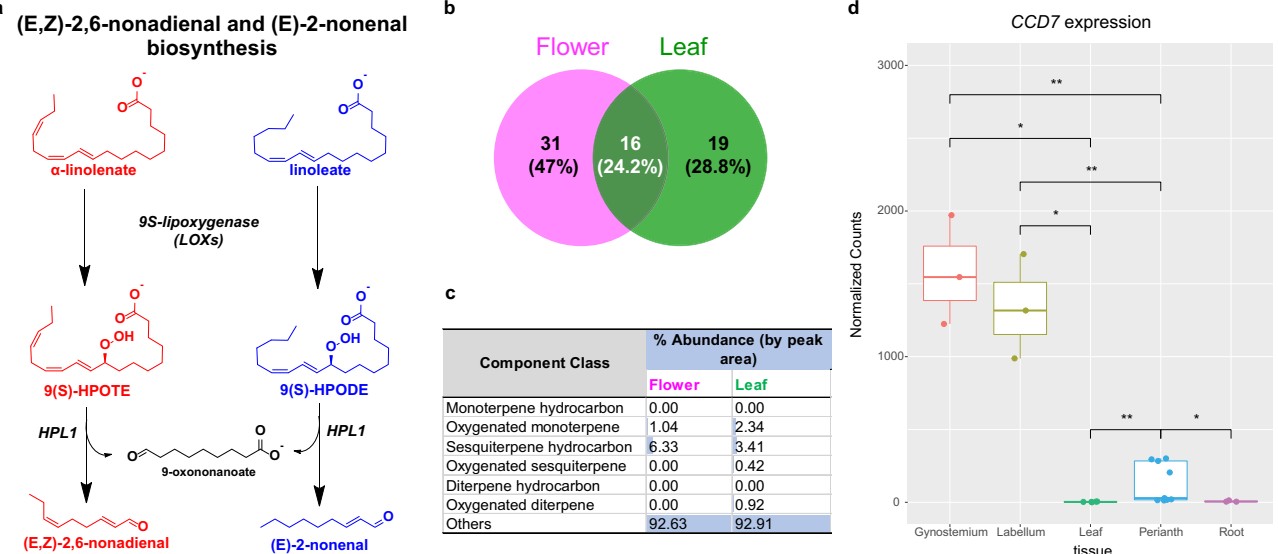

**Fig. 8 Volatile profile of *Ple*. Miss Joaquim 'Agnes' using GC-HRMS. a** Diagram illustrating the biosynthesis of unsaturated C9 aldehydes via two-step enzymatic degradation of linoleate derivatives. **b** Venn diagram showing the overlap between volatile compounds in flower and leaf of *Ple*. Miss Joaquim 'Agnes'. **c** Table shows volatile compounds' abundance by component class and their comparison between flower and leaf tissues. **d** Box plot showing the expression of *CCD7* across the tissues of PMJ. The labellum and gynostemium of PMJ show enhanced activity of *CCD7*, the enzyme responsible for beta-ionone production. For each tissue, $n = 3$ biological independent samples were considered the center line show the medians and box limits indicating the 25th and 75th percentiles. Wilcoxon rank sum test was performed pairwise across tissue groups, and only statistically significant differences across means are reported in the plot. (Source Data: Supplementary Data 19, Supplementary Data 20, *$p <= 0.05$, **$p < 0.01$).

and the compound functions as a protective toxin against potential fungal infection agents[59]. Like pinosylvin, resveratrol possesses antimicrobial properties[60] and thus is likely produced as a defense mechanism against invading microbes. Resveratrol may have potential as an antibiotic enhancer, with demonstrated synergy with polymyxin B against particular multi-drug resistant bacteria[60,61].

*Volatile compound profiling in leaves and flowers of* Ple. *Miss Joaquim 'Agnes'*. PMJ is generally thought to be unscented. However, putative genes in the genome of PMJ encoding for *LIPOXYGENASE* (*LOX*) were identified. *LOX* has been reported to catalyze the many downstream volatiles from a precursor of linoleate or α-linoleate, such as the biosynthesis of major Green Leaf Volatiles constituents, (E, Z)−2,6-nondienal, (E)−2-nonenal[62–64] or methyl-jasmonate through the jasmonate pathway[65,66] (Fig. 8a). This suggests that PMJ may harbor scents from various oxygenated hydrocarbons.

To understand the scent in the hybrid, volatile compound, profiling of freshly grounded flowers and leaves was performed using Solid Phase Micro-Extraction Gas Chromatography-High Resolution Mass Spectrometry (Supplementary Fig. 7). In total, 66 unique compounds from the flower and leaf of PMJ were identified, including (E, Z)−2,6-nondienal (E)−2-nonenal (Supplementary Data 8, 9). 25% of the volatiles were found in both leaves and flowers (Fig. 8b). The abundance of these volatiles remained broadly identical for each component class in the leaves and flowers (Fig. 8c). Notably, low concentrations of floral-associated volatiles such as terpenoids were detected in the flowers (Fig. 8c), consistent with the common observation that the flowers of PMJ do not have a noticeable floral scent.

Low levels of β-ionone, linalool, and vanillin were also detected through GC-HRMS. The biosynthesis of β-ionone is a secondary metabolite of the carotenoid pathway, and its biosynthesis is encoded by *CAROTENOID CLEAVAGE DIOXYGENASE 7* (*CCD7*). *CCD7* is catalyzed through the asymmetrical cleavage of the polyene backbone of β-carotene at C9-C10 position to

β-ionone[67–70]. Transcriptome analysis showed that *CCD7* expression levels were significantly enriched (Log2FC: 6.6, padj: 2.19E-20) in the labellum and gynostemium compared to other tissues of PMJ, indicating that the biosynthesis of β-ionone may occur within the labellum and gynostemium (Fig. 8d and Supplementary Data 10). Collectively vanillin, linalool and β-ionone were among the low concentration floral-associated volatiles detected in the flowers of PMJ. These trace amounts of profiled floral-associated volatiles are likely to form the scent harbored by PMJ (Supplementary Data 10).

## Discussion

Orchidaceae is a highly diverse family of angiosperms, and the genomes of only a handful of its members have been assembled to date. Many bioactive compounds with various pharmacological effects have been extracted from the orchids[6], but many more remain to be discovered. Rather than solely relying on chemical profiling, a complementary and integrative approach of combining genomic and chemical profiling was used here to facilitate the discovery and profiling of phytocompounds and their biosynthesis process.

This study assembles the PMJ genome by combining third-generation sequencing technology for assembly and Hi-C for scaffolding. The improvements in contiguity, accuracy and completeness showed in recently assembled genomes[71] have highlighted the importance of advancing assembly algorithms and sequencing technology to generate high-quality representative genomes.

Traditionally, sequencing studies in plants have focused on agriculturally valuable plants to improve traits, values, and yield. Both genomic and transcriptomics research is essential to land insights into identifying genes in developmental and metabolic pathways. It is exhaustive to comprehensively profile the whole transcriptome involving the various developmental pathways and their metabolic processes. It requires tissue of different developmental stages and many iterations of laborious annotation. Therefore, chemical profiling can be used for phytocompound

screening, including secondary metabolites (i.e., phenylpropanoids, flavonoids, anthocyanins, and stilbenoids) to provide supporting evidence to identify the association between crucial gene involvement, specialized traits, and characterization of secondary metabolites.

The floral colors of PMJ are determined by anthocyanin composition and regulated by genes in the anthocyanin biosynthesis pathway [46]. Anthocyanin derivatives and their composition were assessed in different tissues of PMJ. The expression of *ANS* is consistent with the ion-abundance of anthocyanin pigments measured LC-HRMS in the other floral tissues of PMJ (Fig. 1a). Fragment analysis of the pigments then revealed the various anthocyanin-based colorations in the flowers (Fig. 4). However, fragment analysis is limited by differences in molar absorptivity coefficients and ionization efficiencies of each anthocyanin compound. The absence of chlorophyll demetallation derivatives (pheophytin-a and pheophorbide) in the flowers is consistent with the lack of observed green coloration. Besides coloration, anthocyanins and flavonoids play diverse physiological functions, including developmental regulation, insect attraction, protection against ultraviolet radiation and pathogens, and other signaling pathways[72]. They are also considered valuable bioactive compounds widely recognized for their antioxidative properties[48,73].

Vandaterosides, which are glucosyloxybenzyl eucomate derivatives, were first discovered in *Ple. Teres* stems[49]. In a previous study, human immortalized keratinocyte cell lines (HaCaT) stimulation with eucomic acid and vandateroside II was shown to upregulate cytochrome c oxidase without enhancing mitochondrial biogenesis[74]. As aging is commonly associated with mitochondrial dysfunctions leading to increased oxidative stress and declining cellular energy[75,76], the stimulation of cytochrome *c* oxidase plays a vital role in maintaining mitochondrial functions [77]. Hence eucomic acid and its derivatives can serve as natural compounds with the potential to remedy skin aging.

Green leaf volatiles confer the scent profile of PMJ has been found in several plants, including cucumber (*Cucumis sativus*)[78] and watermelon (*Citrullus vulgaris*)[79]. One common trigger that can activate the production and release of green leaf volatiles is tissue damage. Under such conditions, Green leaf volatiles act as chemical signals for defense response[64]. These two C9 aldehydes, green leaf volatiles, likely gives PMJ a mellow grassy smell (Fig. 8c).

Besides green leaf volatiles, trace amounts of β-ionone, linalool, and vanillin were detected in the flowers, contributing to the scent of PMJ. These volatiles contains a mix of sweet, floral, and woody odors, all highly regarded in perfumery[80,81]. In nature, these compounds function as ecological cues to repel insects like flea beetle (*Phyllotreta Cruciferae*) and butterflies (*Pieris rapae*) [82] while attracting pollinating agents such as orchid bees[80]. Given the visual and specific localization of carotenoids in the labellum of PMJ flowers (Fig. 8d), it is plausible that these carotenoids serve to attract specialist pollinators through β-ionone production. Apart from β-ionone, the degradation of carotenoids by CCDs also yields other important signaling molecules involved in diverse functions like growth development and aroma[68–70].

In conclusion, by integrating and complementing genomic profiling with chemical profiling, insights into the genetic and biochemical basis of the underlying pathways governing the biosynthesis of secondary metabolites were revealed, driving the discovery of associated phytocompounds in the different organs of PMJ. The chemogenomic profiles of the hybrid offer opportunities for further functional studies, serving as an excellent system for understanding the developmental biology and metabolism of orchid plant biology. The draft genome may also contribute to biodiversity conservation and natural heritage preservation.

## Methods

**Plant materials**. PMJ was authenticated and collected from the Singapore Botanic Gardens, National Parks Board (NParks). High Molecular Weight DNA (HMW-DNA) was extracted using Nanobind Plant Nuclei Big DNA Kit (Circulomics Inc)[83]. HMW-DNA was quantified and assessed for purity before using Qubit and Nanodrop. A ligation sequencing kit (LSK109-ONT, Oxford Nanopore Technologies) was used to construct the initial library for sequencing by following the standard protocol provided by the manufacturer. Long read sequencing was performed on GridION using eight r9.4.1 flow cells (Oxford Nanopore Technologies), and base calling was performed using Guppy 4.5.4 (Supplementary Data 11). Illumina paired-end sequencing reads were obtained using an Illumina NovaSeq 6000 from a PCR-free whole genome sequencing library, amounting to 70Gbp of data and 35× coverage of the estimated genome size (Supplementary Data 12).

**Denovo assembly**. Before genome assembly, single-molecule sequencing reads produced by Oxford Nanopore were filtered with QV > 8 and a minimum read length of 1kbp. These filtered reads were then assembled using Flye v2.9[84] with the supplied parameters (—min-overlap 10000). Purge_dups[85] was used to identify and remove duplicated haplotypes from the initial assembly with default parameters. The resulting assembly was then polished using POLCA[86], distributed as part of the MaSuRCA v4.0.3[87] release with illumina whole-genome shotgun sequencing to improve assembly consensus.

**Omni-C library preparation and sequencing**. The Omni-C library was prepared using the Dovetail Omni-C Proximity Ligation Assay (Dovetail Genomics, Scotts Valley, CA, USA) following the manufacturer's protocol (manual version 1.0 for non-mammalian samples). Young meristems of PMJ were harvested, flash-frozen, and homogenized. The chromatin was fixed with formaldehyde in the nucleus, extracted and digested with sequence-independent endonucleases. After digestion, 5′ overhangs were filled with biotinylated nucleotides, and the free blunt ends were ligated. Formaldehyde crosslinks were reversed, and the DNA was purified and treated to remove biotin, not internal to ligated fragments. Finally, sequencing libraries were constructed using Illumina-compatible adapters (Illumina). Biotin-containing fragments were isolated using streptavidin beads before PCR enrichment of each library. The libraries were sequenced using Illumina NovaSeq 6000 to produce ~120 million 2 × 150 bp paired-end reads (Supplementary Data 13).

**Genome scaffolding**. 3D-DNA[88] version 190716 is a software pipeline explicitly designed to correct misjoins and scaffold genome assemblies using proximity ligation data. Paired-end sequencing libraries prepared using Dovetail Omni-C Proximity Ligation Assay were aligned to the *denovo* assembly and preprocessed using the juicer pipeline[89] with the parameters (—early flag -s none) for the 3D-DNA pipeline. The 3D-DNA pipeline was executed with parameters (—repeat-editor-coverage 5), which scaffolds the input assembly, iteratively corrects misjoins and polishes the assembly. Eventually, the contact matrix is visually inspected, and the remaining misjoins are manually curated and resolved using JuiceBox v1.22[89]. TGS-GapCloser v1.0.1[90] was used to perform gap-filling on the finalized genome using nanopore-generated reads, followed by a final round of polishing using POLCA with Illumina libraries prepared from whole genome shotgun sequencing.

**Transcriptome sequencing**. For transcriptome sequencing, total RNA was extracted using DNeasy Plant Kit, QIAGEN from leaves, roots, and floral tissues (petal/sepal, labellum, and gynostemium). In brief, 2 µg of total RNA was processed using the TruSeq Stranded Total RNA with Ribo-Zero for plants (Illumina), followed by sequencing on the Illumina Novaseq 6000 platform in paired-end configuration (Supplementary Data 14).

**Transcriptome assembly**. RNA-Seq reads were pre-processed using trimmomatic v0.39[91] with the following parameters (ILLUMINACLIP:TruSeq3-PE.fa:2:30:10 LEADING:3 TRAILING:3 SLIDINGWINDOW:4:15 MINLEN:36) to remove contaminating sequences from adaptors and sequences with low base quality. RNA-Seq reads were then aligned to the assembled PMJ genome using HISAT2 2.2.1[92] with default parameters. Reference-guided assembly was performed on the aligned RNA-Seq to produce a plant-organ-specific set of transcripts using StringTie v2.2.0[93]. A non-redundant set of transcripts observed across different tissue types was merged using the merge functionality in StringTie using the parameter (-m).

**Genome annotation**. TE and LTR elements were identified from the PMJ genome, and a collective library of these elements was built using RepeatModelerV2.0.3[27]. An BLAST database was built, which is required as input to RepeatModeler with the BuildDatabase utility. RepeatModeler was executed with the following parameters (RepeatModeler -engine 'NCBI' -LTRstruct). Annotation of interspersed repeats and low complexity sequences in the PMJ genome was performed using RepeatMasker 4.1.2-p1[26] with de novo identified repeat libraries of PMJ produced by RepeatModeler using the default parameters. Structural annotation of genes was then performed using MAKER v3.0.3 pipeline[94], which requires RNA-Seq assembled transcripts from the various floral, leaf and root tissues of PMJ and

protein evidence from well-annotated and closely related species (*A. thaliana, O. Sativa, D. catenatum, V. panifolia, C. sinense*) as input. Gene structure from evidence-based annotation was filtered using the following threshold (eAED < 0.3 and protein > 100 bp in length) before training with SNAP[95] and AUGUSTUS[96], subsequently, two rounds of de novo gene prediction and evidence reconciliation with MAKER. The translated protein sequences were subsequently functionally annotated with eggNOG-mapper v2.1.6[97] and the eggNOG database v5[98].

**Transcriptome analysis**. The output of pre-processed RNA libraries where adaptor and low-quality sequences are trimmed. Trimmed reads were quantified onto discovered transcripts from gene models using SALMON[99] with parameters (-lib IU) and converted to feature counts. DEseq2[41] was used to model raw gene counts for gene expression using a negative binomial model and identify differentially expressed genes across various tissues of PMJ with a padj of 0.05 and Log2FoldChange > 1 across all pairwise combinations. An intersection of the genes across all pairwise comparisons was then performed to identify tissue-specific differentially expressed genes.

**Identification of orthologous and MADS-box gene family analysis**. Ortho-Finder v2.5.4[100] was used to identify orthologous gene families in the genome. The phylogenetic tree was resolved based on single-copy orthologous gene families. The MADS-box protein sequences of *A. thaliana* and the appropriate HMMER 3.0[101] profile (PF00319) were used to identify MADS-box transcription factors in the analysis. HMMER 3.0 from HMMtools was used for prediction filtering hits with an E-value threshold of 1e−4. Homology searches were then performed by multiple sequence alignment using MADS-box protein sequences of PMJ, *P. equestris*[18], *A. shenzhenica*[10], *A. thaliana* and *O. sativa* downloaded from NCBI with ClustalW v2.0[102].

**Sample preparation for GC-MS volatile profiling**. Three replicates of pooled fresh plant material (2 g) were flash-frozen with liquid nitrogen and grounded using mortar and pestle before being transferred into a 20 mL amber headspace glass vial and sealed with an air-tight cap. GC method calibrant was prepared by diluting 1 mg/mL of C7-C30 saturated alkanes standard with hexane to a final concentration of 0.1 mg/mL.

**Volatile profiling using GC-MS**. SPME-GC-HRMS analysis was performed using Agilent GC 7890A, coupled with a 7200B GC/Q-TOF System (Agilent Technologies, Palo Alto, CA). Sampling was done automatically using a Combi-PAL autosampler (CTC Analytics, Zwingen, Switzerland). SPME fiber (2 cm, carboxen/DVB/PDMS, 50/30 μm, stableflex 24 Ga, autosampler) was used to facilitate the analysis of small volatiles. An Agilent ALS Syringe, 10 μL, fixed needle, 23–26 s/42/cone, PTFE-tip plunger was used for liquid injection of calibrant solution.

Extraction of volatiles was performed by exposing the SPME needle (40 mm) at an exposure distance of 1.5 cm into the sample headspace at 80 °C for 30 min, with continuous agitation of 200 RPM. Volatiles were desorbed from SPME fiber by direct desorption for 5 min at 250 °C. The injector distance was set at 0.5 cm into the injector. Pulsed splitless inlet mode was adopted with source temperature set at 250 °C. Helium was used as carrier gas at a 1 mL/min flow rate. An HP-5MS column (Agilent 19091S-433) was employed for gas chromatographic separation. The oven program was set at an initial temperature of 60 °C for 5 min, followed by a ramp of 3.3 °C/min to 180 °C for 3 min and, finally, 3.3 °C/min to 270 °C for 3 min for a total run time of 11 min. MS transfer line was kept at 280 °C, and compounds were ionized in EI mode at 70 eV. Quadrupole was left open with a nitrogen collision gas flow of 1.5 mL/min. A mass range of 30–400 *m/z* was acquired for data processing and compound identification.

Liquid injection of saturated alkanes calibrant was performed under similar operating conditions without SPME procedure. Injection volume was kept at 1 μL, and the syringe was washed thrice, before and after injection, with hexane.

According to the vendor's user manual, acquired data were analyzed with Agilent Masshunter - Unknowns Analysis software to detect peaks using a deconvolution algorithm with an area filter of >2e6. Blank subtraction was performed to eliminate false positive hits. A library search was performed against the NIST17 library, with RT match included for match score computation. Best hits were identified and manually curated.

**Sample preparation for LC-HRMS**. Fresh plant material was flash-frozen with liquid nitrogen and ground down with mortar and pestle. Four parts of 90% methanol were added to 1 part of ground biomass. The mixture was vortexed vigorously for 10 min before centrifugation at 10,000 rpm for 5 min. 1200 μL of resultant supernatant was spiked with 400 μL of 2′,7′-dichlorofluorescein (20 μg/mL), which served as the internal standard. Before LC-MS analysis, the sample was filtered through a 0.22 μm polyvinylidene fluoride (PVDF) filter.

**Non-volatile secondary metabolite profiling using LC-MS**. Data were acquired using an Agilent 1290 Infinity II LC System coupled with G6540B QTOF. Waters CORTECS T3 1.6 μm 2.1 mm×100 mm column was deployed and maintained at 40 °C for gradient elution using 0.1% aqueous formic acid as mobile phase A and acetonitrile containing 0.1% formic acid as mobile phase B. The gradient elution program was set as follows: 1% B (0–4 min), 1–70% B (4–17 min), 70–100% B (17–20.5 min), 100% B (20.5–23.9 min), 100–1% B (23.9–24 min), 1% B (24–28 min), with a flow rate of 0.4 mL/min. The sample injection volume was set at 5 μl.

Electrospray ion source was used for compound ionization. It was performed using autoMSMS with the following settings: *m/z* range: 50–1400 Da, gas temperature: 300 °C, gas flow rate: 8 L/min, nebulizer: 35 PSIG, sheath gas temperature: 350 °C, sheath gas flow: 11 L/min, capillary voltage: 3500 V, Nozzle voltage 0 V (positive mode) 2000 V (negative mode), fragmentor 75 V. Agilent ES-TOF Reference Mass solution was used for mass correction (121.0508, 922.0097 for positive ion mode; 112.9855, 1033.9881 for negative ion mode).

**LC-MS data processing and compound identification**. Acquired data were processed using MS-DIAL[103], with guidance from another protocol[104]. Peak detection settings were set to the following: retention time range: 0-22.8 min, MS1 range: 100–1400, MS2 range: 50–1400, centroid parameters: 0.01 (MS1 tolerance) 0.025 (MS2 tolerance). The minimum peak width of 5 and peak height of 1500 were set for peak detection, with mass slice width of 0.1 Da for peak spotting. Adduct detection was set to [M + H]+ or [M−H]− for positive or negative modes. Peaks were exported to MS-FINDER for compound identification[105]. MS-FINDER parameters were set to search through formula prediction and structural elucidation through in silico fragmentation, with a mass tolerance of 10 ppm for MS1 and MS2. Formula calculation was performed under default settings. A cutoff score of 7.0 was applied for structural elucidation, with a spectral match cutoff of 70% being imposed. The compound search was focused on the following databases: FooDB, PlantCyc, NPA, UNPD, KNApSAcK, NANPDB, and COCONUT. Hits were manually curated before being assigned to the respective chromatographic peaks. Molecular networks were generated using the GNPS server[106] and visualized using Cytoscape[107]. Data analysis was performed using JMP®, Version 14. SAS Institute Inc., Cary, NC, 1989–2021.

**Statistics and reproducibility**. Wilcoxon test was used to perform the pairwise comparison of means across highlighted genes in this manuscript. Wilcoxon tests were performed using ggpubr v0.4.0 in R v4.1. Two-tailed *p* values ≤0.05 were considered statistically significant. For each plant tissue, the sample size used was *n* = 3, as stated in the figure captions.

**Reporting summary**. Further information on research design is available in the Nature Research Reporting Summary linked to this article.

# Data availability

BioProject: The Genome of *Ple.* Miss Joaquim 'Agnes' and supporting sequencing datasets (Oxford Nanopore long-reads, Illumina shotgun sequencing, Dovetail Omni-C) used in this study and transcriptome sequencing datasets have been deposited in NCBI Sequence Read Archive under the accession code PRJNA764569. Raw data underlying data shown in the main figures are found in the Supplementary Data. Additional information can be obtained from the corresponding author upon reasonable request.

# Code availability

The parameters used for genome assembly and annotation have been archived in the following Github repository. (https://doi.org/10.5281/zenodo.7043368).

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

## Acknowledgements
This study is jointly funded and supported by the Genome Institute of Singapore and the Duke-NUS Institute of Biodiversity Medicine. The authors recognize the contribution of Verdant Foundation, Chen Lin Trust, Chan Ki, and Yu Tao. The authors would also like to thank National Parks Board, Singapore, for providing and authenticating some of the materials in this study.

## Author contributions
The study was conceived and coordinated by B.T.T., P.T., N.N., K.S., and C.C.Y.N. C.C.Y.N. coordinated and identified material collection. All experiments designed were performed by S.C.C., W.L., and Z.J.L. A.H.L., P.N.S., M.S., and R.V. performed genome assembly and annotation. Data analysis was performed by A.H.L. and Z.J.L. Identification of plant secondary metabolites using mass spectrometry was performed by Z.J.L. The manuscript was drafted by A.H.L., Z.J.L., J.H.H., and B.T.T. All authors contributed to the final manuscript.

## Competing interests
The authors declare no competing interests.
