## [Peer Review File · Communications Biology]

Reviewers' comments:

Reviewer #1 (Remarks to the Author):

1. Why did the authors choose this local plant cultivar for their research? Is this cultivar a model plant or economically important?
2. Statistics should apply to all data, particularly Figures 5, 6 and 8. Also, a significant difference between means should be applied to all the quantitative data.
3. The SE is too high in Figure 6 gynostemium and Figure 8d gynostemium, it usually means that their variation between replicates is too high and consequently causes a non-significant effect.
4. Avoid using first-person writing throughout the manuscript.
5. The authors need to describe their results according to statistics.

Reviewer #2 (Remarks to the Author):

General Impressions:

This manuscript describes a whole genome assembly, phylogenetic analysis and metabolic profiling for *Papilionanthe Miss Joaquim 'Agnes' (PMJ)*, a member of the orchid family. Chemicals produced by members of this family have been investigated for a variety of pharmacological uses providing incentive to investigate how these compounds are synthesized and what other valuable compounds these plants produce. The authors used a combination of long-read, Illumina and Hi-C sequencing to assemble 19 pseudo-chromosomal scaffolds and de novo annotate transcripts. The authors characterize the representation in the genome and expression of specific enzymes related to specific metabolic pathways and profile, anthocyanins, vanillic acid, phenylpropanoids and stilbenoids, and volatile compounds in various tissues.

This article would be of interest to individuals working with PMJ or other members of the orchidaceae family as well as those with interest in the metabolites profiled.

Many of the figures fail to adequately communicate the data because they are too busy, and the text is far too small to read. I recommend simplification of the in-text figures with larger more detailed figures included in the supplement.

The methods lack enough detail for reproducibility. The genome assembly and annotation need more details regarding the methods and the statistics for the libraries used including such details as the number of ONT reads as well as the N50 for the ONT reads and other quality metrics. It is unclear how the Illumina reads were used in the genome assembly, and additional details about those libraries are also needed. For each program used, all input options should be clearly stated not just the name of the tool used.

The circos plots, chromatin contact map, and BUSCO scores (Figure 2, Supplemental Figure 1b, and Supplemental Table 2) suggest that despite some manual curation, there are still some incorrectly placed or oriented scaffolds and that the genome assembly is incomplete. The level of heterozygosity cannot be easily determined from the information provided since the k-mer plot in Supplemental Figure 1a could be interpreted as very highly heterozygous or low heterozygosity. Including a smear plot (also produced from GenomeScope) would clarify this question.

I recommend that a supplemental methods section be added to allow adequate space to include all the necessary experimental details for the genome assembly and other complex experiments. This would allow for additional in text elaboration necessary to properly interpret the results of these experiments.

Line based comments:

72: It is unclear how repetitive regions were specified.

76 "evidence of structural annotation" is unclear

89: This statement lacks context. Has PMJ experienced one or more of these WGD?

113: This statement could only be made if the authors showed that loss of these genes resulted in a change in the flower pigmentation, and the data in Table S6 does not directly address this claim.

114-115: Reference needed.

256: Since PMJ has been described as a hybrid between two other orchid varieties, how inbred were the plants profiled?

266: Include library statistics

266-270: Include all parameters for Flye, Purge_dups, and POLCA

280: Include precise library statistics

289: Include all parameters and a description of how the manual curation was performed.

290: Describe the final round of polishing. What tools and what datasets were used?

292: Provide library statistics and the parameters used for trimmomatic

294-295: List all parameters for HISAT2 and StringTie

303-305: List the criteria used to call differentially expressed genes. What padj? What fold change?

307-313: List all parameters of the tools used in this section

Figures:

Figure 1a: A better visual distinction should be drawn between the larger whole flower image and the inset enlargement. I suggest a white box around the inset image and an additional scale bar within that image.

Figure 1b: A scale bar would enhance this figure.

Figure 2: The gene density track and repeat density track are atypical. They show a distribution pattern opposite to most genomes. This may point to either a problem with this figure, problems with the genome assembly or novel biology which should be further examined. The units and scale of the y-axes are unclear in this image. A heat map might be a better way to display this data. The double peak on the pink track for scaffold 9 suggests there may be problems with this scaffold. Insufficient details are provided in the text of the manuscript to give this data context.

Figure 3: The text on these images is far too small.

Figure 4: The left portion of each panel is too small to see any details.

Figure 5b: Even at high magnification, this figure does not clearly communicate the network relationships. I suggest this should be a separate full-size figure or a supplemental figure.

Figure 5d: The y-axis values seem arbitrarily chosen. They are not uniformly spaced over the range profiled nor are they equidistant from the plot. The bars in this figure appear to have been manually drawn over other bars leading to a concern that the image may have been inappropriately modified. The placement of the blue bar in the Green category is inconsistent with the placement in the other categories. The label is low resolution and too small. The font and spacing of the title are inconsistent.

Figure 6a: The line graph on top of the bar chart is unnecessary and distracting. Please specify what the error bars represent.

Figure 6b: This is a much clearer representation of Ion abundance than 5d. I suggest you model 5d on this figure.

Reviewer #3 (Remarks to the Author):

The authors sequenced the genome of *Papilionanthe* using a combination of short and long-read sequencing and HIC techniques. They further sequenced transcriptome from different tissues of the species followed by chemical (MS) profiling of metabolites from floral and leaf tissues. Identifying and combining the multi-omics information will enrich the Orchids' genomic resources and could favor future research in gene and metabolite engineering as well.

However, following major points should be clarified before the manuscript is published:

1. The quality of figure 5d should be improved
2. Light discussion. The facts presented seems an introduction to the nature and benefits of the detected chemical compounds, not directly corresponding to the theme of the manuscript.
3. No mention or summary of the conclusion of transcriptomics analysis and differential expression from different tissues in the results section

Reviewer #1 (Remarks to the Author):

We appreciate the time and effort that you have taken to carefully examine and provide feedback to improve the manuscript.

Comment	Response
1. Why did the authors choose this local plant cultivar for their research? Is this cultivar a model plant or economically important?	We thank the reviewer for the enthusiasm in our manuscript. Papilionanthe Miss Joaquim 'Agnes' (PMJ) is chosen as a research subject because as the national flower of Singapore, it bears significant horticultural, historical and cultural significance in Singapore. PMJ is one of the first bred hybrid cultivar to gain popularity internationally. Over hundreds of years, it has been used as a breeding stalk for over 440 various cultivars. Favorable characteristic traits such as inducing variable color pallets, fragrance, year round free-blooming, long-lasting flowers, multiple inflorescence and compact growth habits have always been pursued by horticulturalists. These traits were traditionally obtained through breeding or the induction with antimetabolic agents. Hence, studying the PMJ using a multi-omics approach would build resources for orchid breeding and favor genetics and metabolite engineering applications, to benefit the horticultural industry and economically important.
2. Statistics should apply to all data, particularly Figures 5, 6 and 8. Also, a significant difference between means should be applied to all the quantitative data.	We thank the reviewer for the suggestion. We have applied statistical testing using significance between means for Figure 6a and 8d. Wilcoxon rank sum test was performed in a pairwise manner across distribution of means across the various tissue groups. z-score transformed data was also in Figure 5a allows the results to be described statistically.
3. The SE is too high in Figure 6 gynostemium and Figure 8d gynostemium, it usually means that their variation between replicates is too high and consequently causes a non-significant effect.	We thank the reviewer for the detailed reading. We agree with the reviewer that the high SE may indicate that the variation between the replicates in the gynostemium in Figure 6 and Figure 8d is high hence causing a non-significant effect. However, we would like to emphasize that for both figures, the intent is not to compare gene expression across the different tissues. In Figures 6a and 8d, the objective is to show the presence of BADH, which is a possible candidate gene for catalyzing the esterification of Eucomic acid, a vandateroside precursor and

	CCD7, cleaving carotenoids into β-ionone respectively. We have made this clear in line 177 -182 of the manuscript. We have also integrated statistical testing into figure 8d which allow the description our results according to statistics. This can be seen in line 226-228 of the revised manuscript strengthening the conclusion of our results. The conclusion drawn from the results remains unchanged with the addition of statistical testing to the quantitative datasets in Figures 6a and 8d.
4. Avoid using first-person writing throughout the manuscript.	We thank the reviewer for pointing this out. All occurrences of first-person writing in the manuscript have been removed (in lines 12 and 24 of the abstract, in line 235 of the discussion and in line 252 of the conclusion.
5. The authors need to describe their results according to statistics.	We thank the reviewer for the suggestion. In the revised manuscript, Wilcoxon rank sum test have been applied to the datasets in the manuscript and described using statistical methodologies. Please refer to the manuscript lines 177 - 182 and lines 226 – 228 in the updated manuscript.

Reviewer #2 (Remarks to the Author):

General Impressions:

This manuscript describes a whole genome assembly, phylogenetic analysis and metabolic profiling for *Papilionanthe Miss Joaquim 'Agnes'* (PMJ), a member of the orchid family. Chemicals produced by members of this family have been investigated for a variety of pharmacological uses providing incentive to investigate how these compounds are synthesized and what other valuable compounds these plants produce. The authors used a combination of long-read, Illumina and Hi-C sequencing to assemble 19 pseudo-chromosomal scaffolds and de novo annotate transcripts. The authors characterize the representation in the genome and expression of specific enzymes related to specific metabolic pathways and profile, anthocyanins, vandaterosides, phenylpropanoids and stilbenoids, and volatile compounds in various tissues.

This article would be of interest to individuals working with PMJ or other members of the orchidaceae family as well as those with interest in the metabolites profiled.

Many of the figures fail to adequately communicate the data because they are too busy, and the text is far too small to read. I recommend simplification of the in-text figures with larger more detailed figures included in the supplement.

The methods lack enough detail for reproducibility. The genome assembly and annotation need more details regarding the methods and the statistics for the libraries used including such details as the number of ONT reads as well as the N50 for the ONT reads and other quality metrics. It is unclear how the Illumina reads were used in the genome assembly, and additional details about those libraries are also needed. For each program used, all input options should be clearly stated not just the name of the tool used.

The circos plots, chromatin contact map, and BUSCO scores (Figure 2, Supplemental Figure 1b, and Supplemental Table 2) suggest that despite some manual curation, there are still some incorrectly placed or oriented scaffolds and that the genome assembly is incomplete. The level of heterozygosity cannot be easily determined from the information provided since the k-mer plot in Supplemental Figure 1a could be interpreted as very highly heterozygous or low heterozygosity. Including a smear plot (also produced from GenomeScope) would clarify this question.

I recommend that a supplemental methods section be added to allow adequate space to include all the necessary experimental details for the genome assembly and other complex experiments. This would allow for additional in text elaboration necessary to properly interpret the results of these experiments.

We greatly appreciate the time and effort that the reviewer have taken to carefully examine and provide detailed feedback to improve the manuscript. We have addressed the comments through improving manuscript text and figure suggested. The manuscript have also been improved with supporting context and claims with evidence from literature in the results and discussion section. The figures in the manuscript have also been improved so that the results can be described according to text.

The methodological gaps in the manuscript have also been addressed through two means, first we elaborated and added detailed description and parameters on the various assembly and data preprocessing steps and second to guide reproduction a **'Code Availability'** section directed to a GitHub repository containing the execution command, config and parameters.

We also do acknowledge that based on the evidence given that there are still assembled contigs which could still be unplaced, misplaced or misorientated. However the evidence generated by ONT and Hi-C have been exhausted in the current assembly iteration based on the described methodology. Due the highly abundant repeat families increasing the genome complexity future improvements in assembly and scaffolding techniques can potentially push the genome towards perfection.

In Response Figure 1 below, we have included a smudgeplot to clarify the genome heterozygosity, smudgeplot reported a total 59.7 million heterozygous kmer pairs and estimated the minimal number of heterozygous loci to be 2.91 million. The analysis was conducted using a kmer size of 21.

Response Figure 1: A smudgeplot (left) and log10 scaled smudgeplots showing the distribution of heterozygous kmer pairs generated using Illumina reads.

Line based comments	
Comments	Response
72: It is unclear how repetitive regions were specified.	Agreed, the context has been added to line 72 to specify how repetitive regions were determined inside the genome. Repetitive regions in the genome of PMJ were identified and sampled using RepeatModeller to build a library of repeat family de-novo, Repetitive regions in the PMJ genome were then screened and classified using RepeatMasker which is a combination of simple repeats, low complexity sequences and identified repeat families from RepeatModeller. Please see lines 73-78 for the amendments.
76 “evidence of structural annotation” is unclear	The sentence on line 76 has been shifted into the materials and method section as we feel that “evidence of structural annotation” can be better elaborated with the necessary context. The purpose of the structural annotation is to identify the structures of the genome that code for proteins, which is a combination of gene structures, coding regions, and the location of regulatory motifs. Evidence of structure is generated by mapping the various floral and vegetative RNA-Seq libraries to the genome of PMJ. A protein database containing proteins of closely related species queried using blastP with the PMJ genome. This evidence from both proteins and RNA-Seq is termed “evidence of structural annotation” and has been carefully rephrased in the manuscript. Please refer to lines 346 to 351 in the updated manuscript.
89: This statement lacks context. Has PMJ experienced one or more of these WGD?	In line 89 our emphasis WGD has been studied extensively among orchidaceae members. In PMJ, determining WGD using Ks distribution approach shows the absence of WGD evidence. A exponentially decreasing slope only points towards the the presence of small scale duplication (Response Figure 2). However WGD ^{1,2} have been described in phalaenopsis genomes which is in the same Aeridinae subtribe of the Orchidaceae family. .

Response Figure 2: Pairwise analysis of the distribution of Non-synonymous mutations (Ks)

Please refer to lines 90-91 for the changes and reference included.

113: This statement could only be made if the authors showed that loss of these genes resulted in a change in the flower pigmentation, and the data in Table S6 does not directly address this claim.

We agree that the statement is lacking context and have referenced several orchid literature to support the context that the enzymes that are shown in Table S6 in the Anthocyanin biosynthesis pathway are directly involved in floral color pigmentation.

We have revised the text in the manuscript with supporting context and references. Please refer to lines 139-142 in the updated manuscript

114-115: Reference needed.

The specific reference has been added. Please refer to line 145 for the changes made.

256: Since PMJ has been described as a hybrid between two other orchid varieties, how inbred were the plants profiled?

The official variant for this hybrid is known as Papilionanthe Miss Joaquim 'Agnes'. Cited in the book Biology of Vanda Miss Joaquim (Hew et al. 2002) PMJ are only propagated through cuttings or shoot tip/stem culture. However, plants obtained through seeds differ from their parents because of gene recombination, therefore when the intent of propagation is for horticultural purposes, cutting and clonal propagation is the practice.

This being said, it is also possible to determine the inbreeding rate using population-based approaches. Unfortunately, we do not have access to that many individuals as the PMJ used for this work have to be authenticated.

266: Include library statistics

The library statistics have been included in supplementary files. Please refer to Supplementary Table 11 for the changes.

266-270: Include all parameters for Flye, Purge_dups, and POLCA

The workflow and parameters used for Flye, Purge_dups and POLCA have been distributed in a GitHub repository under **Code Availability**.

280: Include precise library statistics	The library statistics for Omni-C have been included in the supplementary files. Please refer to Supplementary Table 12
289: Include all parameters and a description of how the manual curation was performed.	The parameters used have been uploaded to the Github repository for this project and have also been elaborated in the material and methods section. Please refer to lines 325 - 330 in the revised manuscript.
290: Describe the final round of polishing. What tools and what datasets were used?	POLCA was used for the final round of polishing using Illumina-generated reads from the same tissue obtained. Please refer to line 330 in the revised manuscript for clarifications.
292: Provide library statistics and the parameters used for trimmomatic	The library statistics have been included in the supplementary materials and trimming parameters have been detailed in the methods section. Please refer to supplementary table 13 for the changes incorporated.
294-295: List all parameters for HISAT2 and StringTie	The parameters for HISAT2 and StringTie have been detailed in the materials and methods section. Additionally, the workflow used for HISAT2 mapping and StringTie can be obtained from the GitHub repository under Code Availability . Please refer to line 343 in the updated manuscript.
303-305: List the criteria used to call differentially expressed genes. What padj? What fold change?	Differentially expressed genes were compared in a pairwise manner across the leaf and various floral tissues using raw counts quantified using SALMON. DESeq2 was then used to identify differentially expressed genes in a pairwise manner with a padj cut-off of 0.05 and a fold-change of 0.25. An intersect operation was then performed to identify tissue-specific expressions. We have expanded and made a new section for transcriptome and differential expression analysis. Please refer to lines 358 - 364 for the changes in the updated manuscript.
307-313: List all parameters of the tools used in this section	For the comments from lines 290 - 313 the library statistics and parameters have been detailed in the revised manuscript. Please refer to lines 346 – 358 in the updated manuscript.

Figures:	
Comments	Response
Figure 1a: A better visual distinction should be drawn between the larger whole flower image and the inset enlargement. I suggest a white box around the inset image and an additional scale bar within that image.	Thank you, the suggestion has been added to Figure 1a. A white box around the inset with a scale has been added to the image for better visual distinction.
Figure 1b: A scale bar would enhance this figure.	A scale bar has been added to figure 1b.
Figure 2: The gene density track and repeat density track are atypical. They show a distribution pattern opposite to most genomes. This may point to either a problem with this figure, problems with the genome assembly or novel biology which should be further examined. The units and scale of the y-axes are unclear in this image. A heat map might be a better way to display this data. The double peak on the pink track for scaffold 9 suggests there may be problems with this scaffold. Insufficient details are provided in the text of the manuscript to give this data context.	Thank you for your suggestions, we have used a heatmap to display this data. The heatmap has been included in Figure 2 of the revised manuscript.  Figure 2 of manuscript: Circos representation of Ple. Miss Joaquim 'Agnes' A circos plot of genomic features across 19 pseudo-chromosomal assembled scaffolds, from the outer track to the inner track (a) filled line plot of GC content, and heatmap showing (b) gene density, (c) LTR/Copia density and (d) LTR/gypsy density across the assembled scaffolds.

	Referencing several recently assembled genomes from the Orchidaceae family, several gene density peaks can also be observed within a single scaffold³⁻⁶. We do not think that the double peak for the pink track is a result of novel biology. In this project, the experimental design is focused on correlating transcriptomic and metabolite profiling, however, to profile the whole transcriptome requires various tissue along with their associated developmental stage. Hence, despite our extensive characterization of the transcriptome of the mature flowers, leaves and roots lowly expressed transcripts which might be present in the other developmental stage or tissues may be missed out.
Figure 3: The text on these images is far too small.	We have increased the text size of the images in Figure 3.
Figure 4: The left portion of each panel is too small to see any details.	In the left portion of Figure 4 we are trying to depict the number of chemicals which have been discovered in PMJ. We have changed this into a bar-plot which illustrates the proportion of classified and unclassified compounds characterized.
Figure 5b: Even at high magnification, this figure does not clearly communicate the network relationships. I suggest this should be a separate full-size figure or a supplemental figure.	We thank the reviewer for pointing this out. To ensure that the figures clearly communicate the network relationships, in the revised manuscript, the multi-panel Figure 5 is split into two separate full-size figures. The second part for Figure 5 is now presented in a new Supplementary Figure 7.
Figure 5d: The y-axis values seem arbitrarily chosen. They are not uniformly spaced over the range profiled nor are they equidistant from the plot. The bars in this figure appear to have been manually drawn over other bars leading to a concerns that the image may have been inappropriately modified. The placement of the blue bar in the Green category is inconsistent with the placement in the other categories. The label is low resolution and too small. The font and spacing of the title are	We thank the reviewer for the suggestion. The figure has been amended accordingly. A logarithmic scale was used for the y-axis as the values are widely distributed across several orders of magnitude.

inconsistent.	
Figure 6a: The line graph on top of the bar chart is unnecessary and distracting. Please specify what the error bars represent. Figure 6b: This is a much clearer representation of Ion abundance than 5d. I suggest you model 5d on this figure.	In the revised manuscript, the bar chart in Figure 6 and figure 8 have been replaced with a box plot and whisker plot and scatter points to better illustrate the distribution of the datapoints. The clarity of figure 5 have also been improved by splitting the multi panel figure into two separate figures, A new supplementary figure 7 have been created for Figure 5c and d.

Reviewer #3 (Remarks to the Author):

The authors sequenced the genome of *Papilionanthe* using a combination of short and long-read sequencing and HIC techniques. They further sequenced transcriptome from different tissues of the species followed by chemical (MS) profiling of metabolites from floral and leaf tissues. Identifying and combining the multi-omics information will enrich the Orchids' genomic resources and could favor future research in gene and metabolite engineering as well. However, following major points should be clarified before the manuscript is published:

Comments	Response
1. The quality of figure 5d should be improved	Thank you for the comments, to improve upon the quality of figure 5d we have split figure 5 into two separate figures (now figure 5 and Supplementary Figure 7) for clarity improvements.
2. Light discussion. The facts presented seems an introduction to the nature and benefits of the detected chemical compounds, not directly corresponding to the theme of the manuscript.	Thank you for your valuable feedback and we have improved and strengthened the discussion of our manuscript with the following additional key points: 1. The challenges in genome assemblies, fully capturing the diversity of repeat families and gene annotations as well as the importance of orchid genetic resources.2. Using a combination of genetic and chemical profiling to look for special horticultural traits, characterize phytochemicals to advance orchid biology, the study of secondary metabolism and genetics.3. Regulation of traits such as pigmentation, color, rewards, fragrance and their diverse roles in their ecosystem and the limitation of our works.
3. No mention or summary of the conclusion of transcriptomics analysis and differential expression from different tissues in the results section	Thank you for highlighting this and suggesting a component that can be extrapolated from our datasets. We have added a section titled "Transcriptomics analysis from various tissues of PMJ" in the results section to summarize differentially expressed transcriptome analysis derived from various vegetative and floral tissues of PMJ. Please refer to lines 106 - 129 in the revised manuscript.

References

1. Cai, J. *et al.* The genome sequence of the orchid *Phalaenopsis equestris*. *Nat. Genet.* **47**, 65–72 (2015).
2. Chao, Y. *et al.* Chromosome-level assembly, genetic and physical mapping of *Phalaenopsis aphrodite* genome provides new insights into species adaptation and resources for orchid breeding. *Plant Biotechnol. J.* **16**, 2027–2041 (2018).
3. Bae, E.-K. *et al.* Chromosome-level genome assembly of the fully mycoheterotrophic orchid *Gastrodia elata*. *G3 GenesGenomesGenetics* **12**, jkab433 (2022).
4. Xu, Q. *et al.* Chromosome-Scale Assembly of the *Dendrobium nobile* Genome Provides Insights Into the Molecular Mechanism of the Biosynthesis of the Medicinal Active Ingredient of *Dendrobium*. *Front. Genet.* **13**, (2022).
5. Niu, Z. *et al.* The chromosome-level reference genome assembly for *Dendrobium officinale* and its utility of functional genomics research and molecular breeding study. *Acta Pharm. Sin. B* **11**, 2080–2092 (2021).
6. Sun, Y. *et al.* The *Cymbidium goeringii* genome provides insight into organ development and adaptive evolution in orchids. *Ornam. Plant Res.* **1**, 1–13 (2021).

REVIEWERS' COMMENTS:

Reviewer #1 (Remarks to the Author):

In my opinion, it has been revised accordingly.

Reviewer #2 (Remarks to the Author):

Overall the figures and methods are greatly improved. There are still some details lacking regarding the Illumina WGS used to polish the genome assembly.

Specific comments:

Fig 1a. Formatting problems. Additional imbedded labels are visible depending on the program used to open the PDF.

Fig 8a Formatting problems. Overlapping versions of this figure are visible depending on the program used to open the PDF

60, 322. Please clarify which libraries were used for polishing. Was this polishing done using the Hi-C reads, RNA-seq reads or other short-read libraries? If additional libraries were used, what were the library statistics?

83: List the method used to identify orthologous groups here. Orthofinder?

133. For these highlighted genes, it would be informative to state the magnitude and significance of the relative expression between tissues not just higher or lower.

220. what fold expression and padj value?

292, 312, 327. Refer to the appropriate library statistics tables

325. Were these libraries single or paired?

356. A Log₂ fold-change cut-off of 0.25 is very low. Many of the subtle changes detected by this method likely have little biological significance. For identifying genes which define tissue specific expression patterns a higher cut-off (such as log₂ fold-change ≥ 1) should be chosen. This would in turn lead to more meaningful GO-term enrichment.

Reviewer #2 (Remarks to the Author):

Overall the figures and methods are greatly improved. There are still some details lacking regarding the Illumina WGS used to polish the genome assembly.

Response: Thank you for the comments that have improved the quality of this manuscript. The following changes have been made in the manuscript:

1. Details regarding genome polishing have been updated to lines 92 and 331
2. Figures 1 and 8 have been redrawn, and embedded labels are no longer visible.
3. Methodologies have been updated with relevant details.

Response to Specific comments:

Fig 1a. Formatting problems. Additional imbedded labels are visible depending on the program used to open the PDF.

Response: Thank you for your comments. Figure 1a has been redrawn, and the embedded labels should not be present anymore.

Fig 8a Formatting problems. Overlapping versions of this figure are visible depending on the program used to open the PDF

Response: Thank you for the comments. Figure 8a has been redrawn, and the overlapping versions have been removed.

60, 322. Please clarify which libraries were used for polishing. Was this polishing done using the Hi-C reads, RNA-seq reads or other short-read libraries? If additional libraries were used, what were the library statistics?

Response: Thank you for pointing this out. In line 60 and 322, the libraries used to perform genome polishing was Illumina short-read libraries from Whole Genome Sequencing.

Polishing was performed at two stages of genome assembly. First, an initial reference assembly of the genome was assembled de novo by integrating long-range sequencing reads from Oxford Nanopore Technology (ONT) and polished using Illumina short-reads from whole-genome shotgun sequencing to produce the initial assembly (line 92).

Later, TGS-GapCloser v1.0.1 was used to perform gap-filling on the finalized genome using nanopore-generated reads, followed by a final polishing using POLCA with Illumina libraries prepared from Whole genome shotgun sequencing (lines 334-335 and 356).

83: List the method used to identify orthologous groups here. Orthofinder?

Response: Thank you for your comments. Yes, OrthoFinder was used to identify orthologous groups. A total of 32,238 orthologous gene clusters were identified using OrthoFinder. The method has been specified in lines 115-116. Please also refer to line 392 in the Materials & Methods section.

133. For these highlighted genes, it would be informative to state the magnitude and significance of the relative expression between tissues not just higher or lower.

Response: Thank you for your comments. We have included the FC and padj as requested for the highlighted genes. Please see lines 174-180 in the updated manuscript for the changes reflected.

220. what fold expression and padj value?

Response: Thank you for your comments. We have included the FC and padj as requested. Please see lines 256-259 in the updated manuscript for the changes reflected.

292, 312, 327. Refer to the appropriate library statistics tables

Response: Thank you for your comments. The relevant Supplementary Table numbers have been updated on lines 328, 346 and 361.

325. Were these libraries single or paired?

Response: Thank you for your comments. The libraries were sequenced in a paired-end configuration. Please see the following statement, which has also been updated in lines 358-361 of the manuscript.

Total RNA from the leaves, roots and floral tissues (petal/sepal, labellum, and gynostemium) was extracted using DNeasy Plant Kit, QIAGEN for transcriptome sequencing. In brief, 2 µg of total RNA was processed using the TruSeq Stranded Total RNA with Ribo-Zero for plants (Illumina), followed by sequencing on the Illumina Novaseq 6000 platform in paired-end configuration.

356. A Log₂ fold-change cut-off of 0.25 is very low. Many of the subtle changes detected by this method likely have little biological significance. For identifying genes which define tissue specific expression patterns a higher cut-off (such as log₂ fold-change ≥1) should be chosen. This would in turn lead to more meaningful GO-term enrichment.

Response: Thank you for your comments. We have increased the Log₂ fold-change cut-off from 0.25 to 1. Indeed, this filtering criteria has led to a better definition of tissue-specific genesets yielding a more meaningful GO-term enrichment. Please see **Response Figures 1 and 2** below for the updated GO-term enrichment. These are included in **Supplementary Figures 3 and 4** of the revised manuscript.

Please also see lines 143-158 for revised changes in the manuscript.

Pairwise differential expression was performed using DESeq2³² across all tissue combinations, and upregulated genes were compared to identify tissue-specific expressions. Using a Log₂FoldChange greater than 1 as cutoff, 526 genes were found upregulated only in leaves, 64 genes were upregulated only in the labellum, 516 genes were determined to be upregulated only in the root, 65 genes were upregulated only in perianth tissues, and 118 genes were found to be involved in the regulation of the gynostemium. Enrichment of GO terms was used to understand the biological pathways significant in each tissue through tissue-specific genesets. GO pathways enriched in the leaves of PMJ revealed activities related to photosynthesis, chloroplast activity and precursors for generating energy/metabolites. In the roots, GO terms enriched mainly belong to response to water and production of secondary metabolites, which could play crucial roles in the uptake of nutrients and water from the air and their surroundings (Supplementary Fig. 3). In the floral tissues, gene-set enrichment specific to the perianth shows high specificity towards the regulation of anthocyanin-based metabolism mainly involved in floral pigmentation. GO terms enriched in labellum tissues shows gene-set enrichment involved in the biosynthesis of many secondary metabolites, which could be used to attract pollinators and GO terms enriched in the gynostemium show processes involved in regulating floral development (Supplementary Fig. 4). Therefore, the identification of differentially expressed genes and their enriched pathways demonstrates the diverse biological processes occurring in the various tissues of PMJ that allow each tissue to play specialized roles in growth and development.

Response Figure 1: Enriched GO terms in the leaf and root of PMJ.

Response Figure 2: Enriched GO terms in the floral tissues of PMJ.